# In operando cryo-STEM of pulse-induced charge density wave switching in TaS$_2$

James L. Hart ®[1], Saif Siddique ®[1], Noah Schnitzer[1], Stephen D. Funni[1], Lena F. Kourkoutis ®[2,3] & Judy J. Cha[1] ✉

The charge density wave material 1T-TaS$_2$ exhibits a pulse-induced insulator-to-metal transition, which shows promise for next-generation electronics such as memristive memory and neuromorphic hardware. However, the rational design of TaS$_2$ devices is hindered by a poor understanding of the switching mechanism, the pulse-induced phase, and the influence of material defects. Here, we operate a 2-terminal TaS$_2$ device within a scanning transmission electron microscope at cryogenic temperature, and directly visualize the changing charge density wave structure with nanoscale spatial resolution and down to 300 μs temporal resolution. We show that the pulse-induced transition is driven by Joule heating, and that the pulse-induced state corresponds to the nearly commensurate and incommensurate charge density wave phases, depending on the applied voltage amplitude. With our in operando cryogenic electron microscopy experiments, we directly correlate the charge density wave structure with the device resistance, and show that dislocations significantly impact device performance. This work resolves fundamental questions of resistive switching in TaS$_2$ devices, critical for engineering reliable and scalable TaS$_2$ electronics.

1T-TaS$_2$ is a layered, two-dimensional (2D) quantum material which undergoes an insulator-to-metal transition induced by voltage pulses (Fig. 1a, b)[1–4]. The switching is fast, energy efficient, and reversible, making TaS$_2$ attractive for device applications[5,6]. Moreover, the layered structure of TaS$_2$ may enable atomically-thin memristive or neuromorphic devices, providing ultimate scalability inaccessible to 3D crystals[7]. Nevertheless, knowledge of the switching mechanism in TaS$_2$ is limited. Prior works indicate that the electrical switching is associated with the charge density wave (CDW) structure (Fig. 1b, c)[1–4,8–12]. At low-temperature (<200 K), TaS$_2$ exhibits the commensurate (C) CDW phase, which is insulating[12–14]. At higher temperature, several metallic CDW phases exist[12–14], as well as non-thermal CDW phases accessible with optical excitation[15,16]. Direct characterization of the CDW structure during device operation is limited. In operando scanning tunneling microscopy studies have visualized the CDW structure before and after switching[11,17], but scanning probe studies do not possess the time-resolution to capture the switching process,

and are strictly surface sensitive. TaS$_2$ switching has also been studied with in operando optical measurements[8], but such measurements lack nanoscale spatial resolution, and only indirectly probe the CDW phase. Hence, our understanding of the bias-induced phase(s) is still unclear, and more importantly, the switching mechanism remains unknown. Most studies argue that the transition is field-induced (non-thermal), although there are competing proposals for the microscopic mechanism[1–6,8]. More recently, several groups have claimed that Joule heating is partially or wholly responsible for switching based on finite element simulations[9,10], as well as IR thermal imaging of a bulk crystal under constant bias[18]. A concrete understanding of the switching mechanism is critical for the development of TaS$_2$ electronics. To this end, in operando measurements are needed to correlate the CDW structure, flake temperature, and electrical resistance of a nanoscale device.

Here, we operate a 2-terminal TaS$_2$ device within a scanning transmission electron microscope (STEM) at cryogenic temperature.

[1]Department of Materials Science and Engineering, Cornell University, Ithaca, NY, USA. [2]School of Applied and Engineering Physics, Cornell University, Ithaca, NY, USA. [3]Kavli Institute at Cornell for Nanoscale Science, Cornell University, Ithaca, NY, USA. ✉e-mail: jc476@cornell.edu

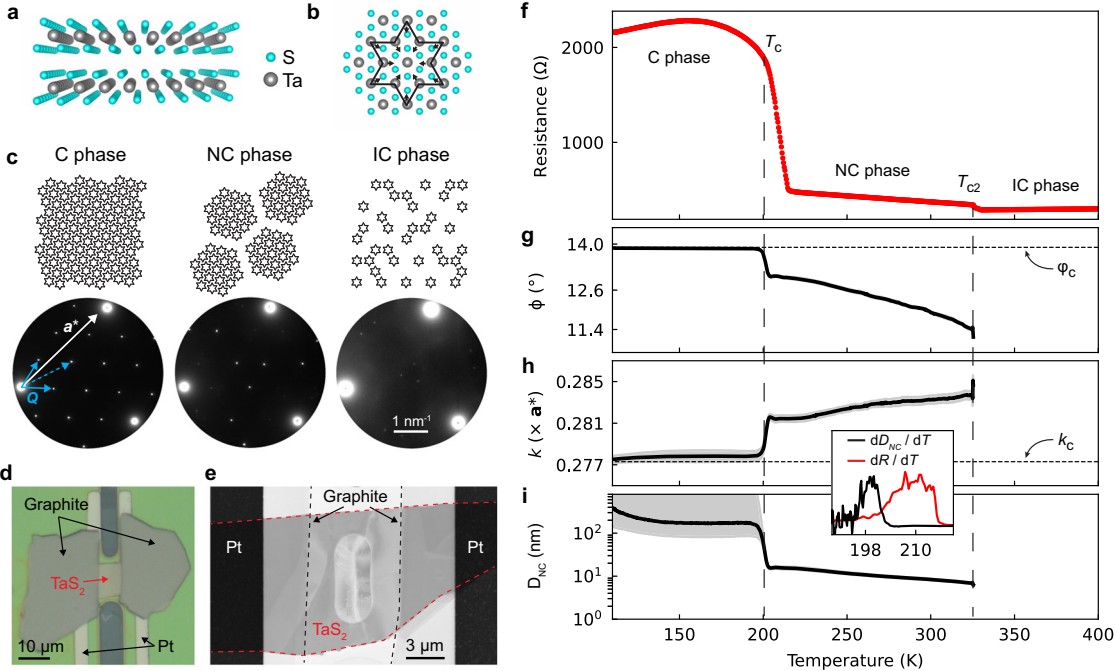

**Fig. 1 | TaS₂ structure and temperature dependent CDW behavior.** Atomic structure of 1T-TaS₂ in cross-section (**a**) and plan-view (**b**). In (**b**), the local CDW distortion is shown, which forms a Star-of-David structure. **c** Illustration of the C, NC, and IC CDW phases, which exhibit different orderings of the stars. Associated electron diffraction patterns are shown. For the C phase diffraction pattern, the Bragg vector **a\*** is shown, as well as two 1st order CDW **Q** vectors (solid line) and one 2nd order CDW vector (dotted line). The studied device is imaged optically (**d**) and with STEM high angle annular dark field imaging (**e**). In the center of the STEM image is a through-hole in the SiN$_x$ membrane, which allows for electron diffraction measurements. **f** Temperature dependence of the TaS₂ resistance. **g**–**i** CDW angle φ, CDW wavevector magnitude $k$, and the domain size $D_{NC}$, respectively. The shaded regions represent the standard error. The inset shows the temperature derivatives of the resistance and $D_{NC}$. For the inset, the $x$-axis units are temperature (K). The temperature-dependent diffraction data is shown in Supplementary Movie 1. The electrical biasing set-up and details are described in Supplementary Note 1.

Through time-resolved electron diffraction and 4D-STEM imaging, we quantify the CDW order parameter during electric biasing, and, via strain analysis, we measure the local sample temperature. By directly correlating the CDW structure, flake temperature, and device resistance during switching, we unequivocally show that Joule heating drives the switching process, both for steady-state bias and short voltage pulses. Accordingly, the bias-induced phases correspond to thermal CDW states. We also show coupling between the CDW order parameter and the device resistance, and we demonstrate how local microstructural features (dislocations) influence device operation. These findings are crucial for the engineering and optimization of TaS₂ devices for beyond-silicon technology.

## Results

The studied device is shown in Figs. 1d,e with optical and STEM imaging, respectively. A bulk 1T-TaS₂ crystal was exfoliated in an Argon glove box onto a SiO₂ / Si substrate, and graphite electrodes were placed on a ≈ 55 nm thick TaS₂ flake (channel length = 7 μm). The finished device was transferred to an in situ TEM chip, and placed over a through-hole drilled in an amorphous SiN$_x$ membrane. The TEM chip has Pt electrodes for in situ electric biasing (Supplementary Note 1), as well as a Pt coil that allows local sample heating and thermometry from ≈100–1000 K[19,20].

We use electron diffraction to characterize the CDW state. The TaS₂ CDW follows the Star-of-David distortion, wherein 13 Ta atoms bunch together (Fig. 1b). In the insulating low-temperature C phase, these stars form a long-range lattice commensurate with the atomic structure[13,14]. As shown in Fig. 1c, electron diffraction of the C phase yields sharp 1st and 2nd order CDW satellite peaks. The C phase CDW wavevector has an angle of φ$_c$ = 13.9° relative to the Bragg wavevector **a\*** and a magnitude of $k_c$ = 0.2773a\*. Above ≈200 K, defects in the CDW

known as discommensurations form and organize into a hexagonal network with a CDW domain size of order ≈10 nm[13,14]. This is the metallic nearly commensurate (NC) phase, which suppresses the 1st order CDW peak intensity, and slightly adjusts the CDW wavevector, with φ ≈ 11–13° and $k$ ≈ 0.280a\*–0.285a\*. Thomson et al. showed that the values of φ and $k$ determine the domain size of the NC phase, $D_{NC}$, according to

$$D_{NC} = a \left/ \sqrt{\left(\frac{2\pi\Delta\varphi}{360}\right)^2 + \left(\frac{\Delta k}{k_C}\right)^2} \right. \tag{1}$$

where $a$ is the atomic lattice parameter, and $\Delta\varphi$ and $\Delta k$ are the differences in φ and $k$ relative to their commensurate values[14]. Here, we use $D_{NC}$ as an order parameter to track the C to NC phase transition, where the order parameter represents a measurable quantity which distinguishes the two CDW phases; $D_{NC}$ ≈ 10 nm within the NC phase and $D_{NC} \to \infty$ within the C phase. Above ≈325 K, TaS₂ transitions to the incommensurate (IC) phase, with φ ≈ 0° and $k$ ≈ 0.285a\*, and Eq. 1 is no longer applicable.

We first study the CDW behavior as a function of temperature. Figure 1f shows the flake resistance upon heating at 0.4 K / s; the C to NC and NC to IC transitions are clearly observed. In total, the resistance drops by a factor of 8. Using a direct detection camera[21], we collect temperature-resolved selected area electron diffraction data simultaneous with the resistance measurements (Supplementary Movie 1). From the diffraction data, we quantify the CDW structure based on the 2nd order CDW spots (Supplementary Note 2). The CDW angle φ and magnitude $k$ are plotted in Fig. 1g, h, and the calculated $D_{NC}$ is shown in Fig. 1i. In the low-temperature C phase, $D_{NC}$ ≈ 300 nm. The large $D_{NC}$ error bars in the C phase reflect the nature of Eq. 1; as $\Delta\varphi$ and $\Delta k \to 0$, small errors in φ and $k$ are magnified in the propagated $D_{NC}$ error.

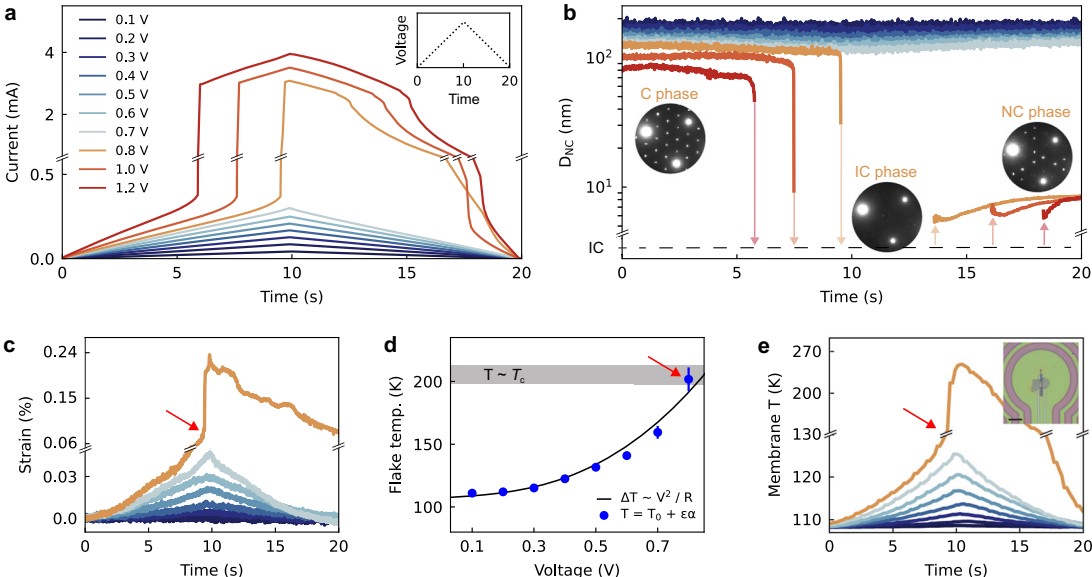

**Fig. 2 | Steady-state biasing and CDW switching. a** Current versus time during triangular voltage ramps with the maximum voltage ranging from 0.1 V to 1.2 V. The maximum voltage is reached at 10 s (see Supplementary Note 1 for biasing details). Inset: example voltage profile. The color legend in (**a**) applies to (**b**, **c**, **e**). **b** The measured CDW domain size $D_{NC}$ during the voltage ramps. The insets show diffraction snapshots acquired during the 0.8 V ramp (Supplementary Fig. 1 provides the full diffraction patterns). The full diffraction datasets for 0.7 V and 0.8 V ramps are shown in Supplementary Movies 2 and 3. **c** Flake strain during the voltage ramps. The statistical error in the relative sample strain measurements is 0.0014% (Supplementary Note 3). **d** Maximum flake temperature for voltage ramps from 0.1 to 0.8 V, calculated from the strain shown in (**c**). For the 0.8 V datapoint, we show the temperature immediately prior to the C to NC transition. $T_0$ is 110 K, ε is the strain, α is the coefficient of thermal expansion, and $R$ is the flake resistance. Error bars reflect the uncertainty in α. The shaded gray region marks the range of $T_c$ values measured upon different warming experiments. **e** Measured temperature of the SiN$_x$ membrane during the voltage ramps. The thermometer consists of a Pt coil encompassing the flake, pictured in the inset. Scale bar is 50 μm.

Upon entering the NC phase at ≈200 K, $D_{NC}$ quickly falls to ≈12 nm, and then gradually decreases to 8 nm before the flake enters the IC phase. By plotting the derivates $dD_{NC} / dT$ and $dR / dT$ (inset), we see that the structural CDW transition precedes the resistive transition by ≈10 K. This finding was only possible given our in operando multimodal experimental approach. As we show later, this result is relevant to device operation.

Next, we study bias-induced CDW switching by applying triangular voltage ramps (20 s duration, Fig. 2a inset) while collecting diffraction patterns at a rate of 100 Hz. These slow ramps effectively probe the steady-state CDW response to an applied electric field. Measurements are performed at ≈110 K, starting in the C phase. Application of triangular voltage ramps with maximum voltages ranging from 0.1 to 0.7 V yield triangular current versus time curves, perfectly reflecting the input voltage profile. This response indicates no electrical switching (Fig. 2a). Within this voltage range, electron diffraction measurements show that the flake remains in the C phase, with minimal changes in the CDW domain size $D_{NC}$ (Fig. 2b and Supplementary Movie 2). In contrast, for voltages above 0.8 V, there is a sudden increase in current, indicating resistive switching. Concurrently, $D_{NC}$ rapidly falls, indicating the C to NC transition, quickly followed by the NC to IC phase transition (Fig. 2b insets, Supplementary Movie 3 and and Supplementary Fig. 1). As the applied voltage decreases on the second half of the voltage ramp, the flake recovers to the NC phase (Fig. 2b), and then slowly progresses towards the C phase over the next several minutes. For this device, the steady-state switching threshold is ≈0.8 V. While the measured current versus voltage behavior is consistent with data in the literature[1–4], our in operando experiment directly quantifies the CDW order parameter (*i.e.* $D_{NC}$) during switching, and identifies the NC and IC phases as the bias-induced states.

We now demonstrate that the bias-induced switching is driven by Joule heating. To measure the local flake temperature during bias,

we first extract the in-plane flake strain ε from the diffraction data (Fig. 2c, Supplementary Note 3). For the sub-threshold voltage ramps of ≤0.7 V, the strain versus time profiles rise and fall with the applied voltage ramp, with larger voltages leading to increased strains. Conversely, for the 0.8 V ramp, there is a rapid strain jump at the bias-induced CDW transition. Next, we convert the strain data to temperature using the thermal coefficient of expansion α for this device (Supplemental Note 3)[22,23]. Note that this method is only applicable when the flake is in the C phase (where $\Delta T = \alpha \varepsilon$ is valid), since the C to NC CDW transition causes a discontinuous lattice change due to CDW-lattice coupling. Figure 2d plots the maximum flake temperature during each voltage ramp based on the strain data shown in Fig. 2c. For the 0.8 V curve, we plot the temperature immediately prior to the CDW transition. The data shows that with biasing at 0.8 V, the flake temperature approaches 200 K, which is the C to NC phase transition temperature (Fig. 1f). Red arrows mark $T_{flake} \approx T_c \approx 200$ K in Figs. 2c, d. Thus, our data clearly shows that at the threshold voltage of 0.8 V, Joule heating is sufficient to raise the flake temperature to $T_c$ and thermally trigger the C to NC transition. Supporting this claim, the temperature versus voltage data is well fit by $\Delta T \propto P = V^2 / R$, where $P$ is the power generated from Joule heating, $V$ is the maximum applied voltage, and $R$ is the flake resistance prior to switching.

Joule heating also explains the rapid rise in strain (and temperature) after switching at 0.8 V: as the CDW transition begins and the resistance drops, the power generated increases according to $V^2 / R$, which further increases the flake temperature and accelerates the transition. This positive feedback loop leads to sudden and complete CDW switching, and, consequently, a spike in the flake temperature and strain. The inherent lattice expansion at the C to NC phase transition[22,24] also contributes to the strain jump during the 0.8 V ramp. This effect results in a remnant positive strain after the voltage ramp is complete, since the flake remains in the NC phase and does not

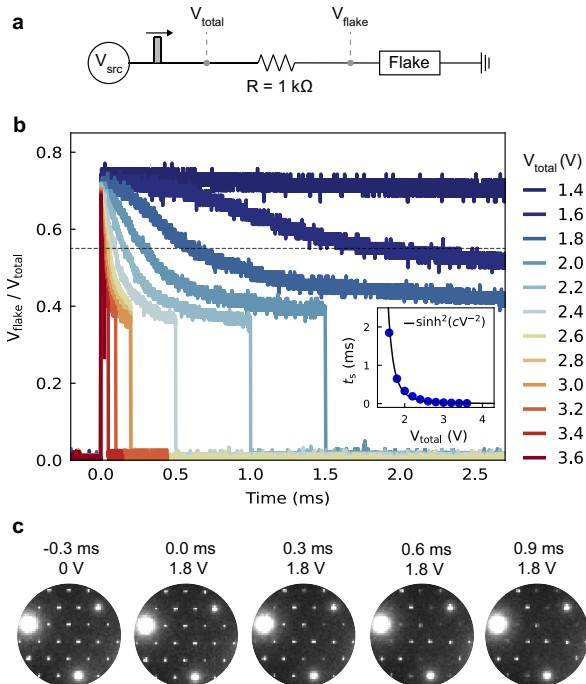

**Fig. 3 | Time-dependent resistive switching. a** Our biasing setup, where $V_{src}$ is the voltage source used to generate square voltage pulses, and $V_{total}$ and $V_{flake}$ are measured with an oscilloscope (see Supplementary Note 1 for details). **b** The measured ratio of $V_{flake} / V_{total}$ for $V_{total}$ ranging from 1.4 up to 3.6 V. We define the switching time $t_s$ when $V_{flake} / V_{total}$ drops below 0.55, as shown with the horizontal dotted line. The inset plots $t_s$ versus $V_{total}$, along with a fit for Joule heating on a 2D membrane provided in Ref. 27. **c** In operando diffraction snapshots from the 1.4 V pulse. The voltage pulse is applied at 0 ms, although there is an uncertainty of ≈0.3 ms in the diffraction pattern timestamps. The diffraction data for this pulse is shown in Supplementary Movie 4, and further analysis is provided in Supplementary Fig. 2.

fully relax to the C phase within the measurement time frame (Fig. 2c, 0.8 V curve).

To provide an independent confirmation of Joule heating, we plot the membrane temperature during the voltage ramps, as measured with the Pt coil on the TEM chip (Fig. 2e). The membrane temperature shows the same qualitative behavior as the flake strain and temperature measurements (Figs. 2c, d), supporting the presence of Joule heating. For the 0.8 V ramp, prior to the CDW transition, the membrane temperature is ≈130 K. Using a simple thermal transport model, we find that a membrane temperature of 130 K is consistent with a flake temperature of 200 K (Supplementary Note 4). We also highlight that after the CDW transition, the membrane temperature shows a temperature spike of >100 K, which supports the positive feedback loop between Joule heating and the insulator-to-metal transition.

It follows from the Joule heating hypothesis that the time needed for switching $t_s$ should scale inversely with the applied voltage[25,26]. Specifically, a model from Fangohr et al.[27] predicts $t_s \propto \sinh^2(cV^{-2})$ for a nanodevice on a 2D membrane, where $c$ is a constant. To evaluate this behavior, we measure the time-resolved resistance during switching using the biasing setup shown in Fig. 3a. We pass square voltage pulses (durations from 3 ms down to 100 μs) through a 1 kΩ resistor in series with the flake, and we plot the voltage drop across the flake divided by the total applied voltage, $V_{flake} / V_{total}$. This ratio scales with the flake resistance; hence, a drop in $V_{flake} / V_{total}$ indicates bias-induced switching (Fig. 3b). There is no resistive switching for a 3 ms 1.4 V pulse, but all pulses >1.4 V initiate switching, with progressively shorter switching times for larger voltage amplitudes. Consistent with the biasing data, in operando diffraction measurements show switching

from the C to the NC phase (Fig. 3c and Supplementary Movie 4). Moreover, the time-resolved diffraction shows that an increase in flake strain (and thus temperature) precedes the CDW transition, as expected from Joule heating (Supplementary Fig. 2). The Fig. 3b inset plots $t_s$ as a function of $V_{total}$, and the $t_s \propto \sinh^2(cV^{-2})$ model provides an excellent fit to the data. This result further confirms the Joule heating-induced switching mechanism. Conversely, for a non-thermal voltage induced mechanism, one would expect rapid (~ps) switching for all voltages above the threshold value[5,6], in clear contrast with our results. The $\sinh^2(cV^{-2})$ fit also suggests that Joule heating can induce switching on the ns timescale, given a sufficiently large voltage pulse. Indeed, many literature reports of pulse-induced switching of $TaS_2$ can reasonably be accounted for via Joule heating[1–4].

We next study coupling between the CDW order parameter and the device resistance through a series of short voltage pulses which are relevant for device operation. We apply pulses with $V_{total}$ starting at 2.0 V and increasing by 0.4 V up to 9.6 V, all with a 3 μs pulse duration, performed at 110 K. Figure 4a shows the measured $V_{flake} / V_{total}$ for a representative set of pulses. Note that the $RC$ time constant for this device is τ ≈ 460 ns (likely due to poor impedance matching throughout the in situ TEM set up), which places an upper limit on device operation speed. Partial switching is observed for $V_{total} \geq 3.2$ V, and full switching is observed for $V_{total} \geq 5$ V. This behavior is consistent with Joule heating and the $t_s \propto \sinh^2(cV^{-2})$ model, which predicts $t_s = 1.1$ μs for $V_{total} = 5.2$ V.

The pulse-dependent evolution of the CDW structure and device resistance is shown in Fig. 4b, c. Several interesting trends are present in the data. First, as the pulse amplitude increases, the pulse-induced $D_{NC}$ decreases, as does the device resistance. Thus, higher voltage pulses produce smaller CDW domains (thus a higher density of discommensurations), which in turn reduce the flake resistance. This behavior is captured in Fig. 4d, which plots the $D_{NC}$ and the flake resistance immediately after pulsing. This finding is consistent with Joule heating, with higher voltage pulses producing larger temperature changes. Comparing the $D_{NC}$ versus $V_{total}$ and resistance versus $V_{total}$ curves in Fig. 4d, we see that the structural CDW transition precedes the electronic resistance transition. This trend is consistent with our earlier finding, that with heating through the C to NC transition, the $D_{NC}$ transition precedes the insulator-to-metal transition (Fig. 1i inset). Secondly, we find that after switching the device recovery is not complete, i.e. the switching is not fully reversible back to the C phase. This is evident in Fig. 4b, c; both the pre-pulse $D_{NC}$ and the device resistance steadily decrease after successive pulsing. We note that after the full pulsing experiment, the CDW structure and resistance were fully reset via heating to the IC phase and then cooling to 110 K.

The partial irreversibility of pulse-induced switching suggests CDW pinning on local microstructural features. Indeed, there are many $TaS_2$ biasing experiments which suggest increased CDW pinning in thin flakes[3,4,8], although direct confirmation is lacking. To test this hypothesis, we performed real-space analysis using 4D-STEM imaging. With this method, the electron beam is focused to a nanoscale probe, rastered across the sample surface, and a full diffraction pattern is captured at each spatial coordinate[28]. From the diffraction patterns, we extract the CDW angle φ and magnitude $k$ using the same method applied to the selected area diffraction data shown in Fig. 1g, h. In turn, the $D_{NC}$ can be mapped in real space with a spatial resolution of ≈20 nm (see Supplementary Note 2 for details). For these measurements we study a separate flake, imaged optically in the Fig. 5a inset. While the flake is a high-quality single crystal, we observe the presence of basal dislocations[29–31], which are revealed with a virtual-STEM image based on Bragg diffraction contrast (see the dark lines in Fig. 5a). We find that all exfoliated $TaS_2$ flakes (over a dozen were observed by STEM) exhibit similar dislocation structures. Figure 5b maps $D_{NC}$ as a function of applied bias at a temperature of 120 K. Initially, the $D_{NC}$ is mostly >50 nm, indicating a spatially homogenous C phase, and the

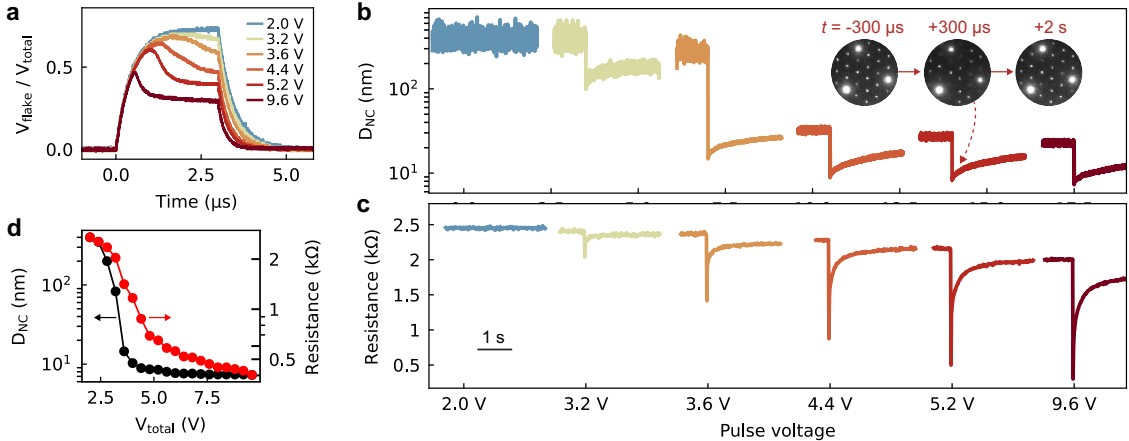

**Fig. 4 | Pulse induced CDW and resistive switching. a** $V_{total}$ / $V_{flake}$ during electric pulsing. Twenty consecutive pulses were performed in total, starting at 2.0 V and increasing by 0.4 V up to 9.6 V, with roughly 5 mins recovery time in between pulses. A representative set is shown here. See Supplementary Note 1 for biasing details. Time-resolved CDW domain size $D_{NC}$ (**b**) and device resistance (**c**) during pulsing. The scale bar shows 1 s. The measurement time resolutions are 300 μs for the CDW analysis and 12 ms for the device resistance. The diffraction data for the 9.6 V pulse is shown in Supplementary Movie 5. **d** Comparison of the CDW domain size $D_{NC}$ (black) and the flake resistance (red) immediately after pulsing.

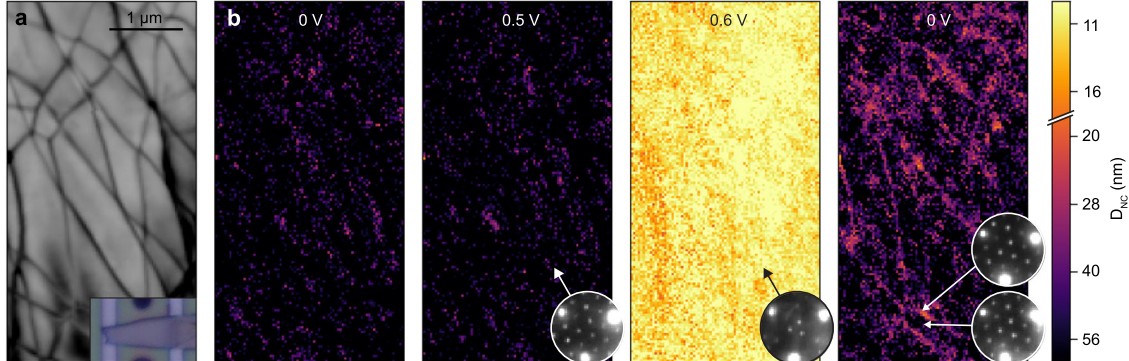

**Fig. 5 | Real-space CDW imaging during bias. a** Virtual STEM image which sums all of the Bragg peak intensities. The dark lines are basal dislocations. The inset is an optical image of this flake. **b** Maps of the CDW $D_{NC}$ as a function of applied bias. The insets show cropped diffraction patterns, extracted from local regions of 3 × 3 pixels. For the post-bias dataset (far right), the top diffraction pattern is extracted from a dislocation, and the bottom diffraction pattern is extracted from a non-defective region.

device is in a high resistance state (resistance = 2.11 kΩ). With application of 0.5 V, the CDW remains in the C phase, and the device resistance remains high (1.99 kΩ). At 0.6 V, the device switches; the CDW map shows $D_{NC} \approx 10$ nm, and the resistance drops to 464 Ω. Note that this behavior is similar to our results in Fig. 2, albeit with a slightly lower threshold voltage, and the NC phase is more stable relative to the IC phase (for this flake, 0.8 V causes a transition to the IC phase). After releasing the applied bias, the sample resistance increases to 1.8 kΩ, and the CDW map mostly shows $D_{NC} > 50$ nm. However, the NC phase is found to locally persist at the basal dislocations, with $D_{NC} \approx 30$ nm adjacent to the line defects. This data suggests that after pulsing, discommensurations of the NC phase are pinned to dislocations, preventing complete recovery to the C phase. This real-space analysis provides a microscopic understanding of CDW pinning in TaS₂ devices.

## Discussion

By operating a 2-terminal TaS₂ device within the STEM at cryogenic temperature, we demonstrate that bias-induced switching is driven by Joule heating and a rapid thermal transition from the C to the NC and IC phases. In our device, this mechanism is operative for both steady-state biasing and μs voltage pulses. Our data also indicates that Joule heating can drive switching on the ns timescale. Based on this

knowledge, we suggest engineering heat management of TaS₂ devices, e.g. the thermal conductivity of the substrate and the electrode geometry, in order to efficiently reach $T_c$ with minimal losses. While most reports of TaS₂ switching are consistent with Joule heating, we note recent claims of picosecond switching, with a switching energy seemingly below the heat needed for Joule heating[25,26]. Hence, it may be that under certain conditions, purely field-induced switching is possible. Our finding that dislocations can pin the CDW structure and prevent complete recovery is relevant to the long-term reliability of TaS₂ devices under continuous operation. Devices could also be engineered with optimized dislocation structures to help stabilize the NC phase after switching, which could prolong the lifetime of the pulse-induced low-resistance state. Dislocation engineering may also enable faster switching as well as multi-level resistance states. Here, we studied a relatively thick ≈55 nm flake, and further work is needed to understand the Joule heating mechanism and dislocation pinning in thinner flakes, which show distinct CDW behavior[3,4] and may also possess distinct defect structures[32]. Further work is also needed to understand interfacial effects, particularly for thin samples[3,33]. For instance, encapsulation with hexagonal BN can strongly influence surface oxidation and strain[3,34], though how this influences the CDW remains poorly understood. Lastly, our use of in operando biasing and cryogenic cooling, along with 4D-STEM imaging of the CDW order parameter,

offers a promising route to understand device performance for other quantum materials[35].

## Methods

The bulk 1T-TaS$_2$ crystal was purchased from 2D Semiconductors. For our in operando (S)TEM experiments, we used a dual-tilt cryogenic holder from HennyZ with 6 biasing pins (model FDCHB-6), allowing for variable temperature operation and sample biasing. We used heating and biasing nanochips from DENS solutions (part # DS-P.T.2B4H.DS-1). In situ electric biasing was performed with a Keithley 2400 SMU, a Keysight 33600 A waveform generator, and a Tektronix DPO2024 oscilloscope. Details regarding our electrical measurements are provided in Supplementary Note 1. (S)TEM measurements were performed with a Thermo Fisher Titan Themis 60-300 kV instrument, operated at 120 kV. Both the time-resolved diffraction (Figs. 1–4) and the 4D-STEM (Fig. 5) datasets were captured using an EMPAD-G2 detector. For the selected area diffraction measurements, the beam current was ≈2 nA and the beam size was ≈2 μm (Supplementary Fig. 3). For the 4D-STEM mapping, the beam current was ≈1.5 nA, and the convergence semi-angle was set to 0.15 mrad, providing a nominal real-space probe size of ≈14 nm (Supplementary Fig. 4). The pixel dwell time was 2 ms and the step size was 21 nm. Electron energy-loss spectroscopy was used to estimate a sample thickness of 55 nm. The measured thickness was 0.9 mean free paths, and we used an inelastic mean free path of 61 nm for TaS$_2$ at 120 kV, as calculated using David Mitchell's plugin from www.dmscripting.com. Data processing steps are described in detail in Supplementary Note 2 and Supplementary Figs. 5–8.

## Data availability

The electron diffraction datasets used for strain and temperature analysis are available at DOI: 10.34863/wgf1-pw79.

## Code availability

The code used for strain analysis is available at https://doi.org/10.34863/wgf1-pw79.

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

## Acknowledgements

J.L.H. and J.J.C. were funded through the Gordon and Betty Moore foundation (EPiQS Synthesis Award). S.S. acknowledges funding from DOE BES DE-SC0023905. N.S. acknowledges support from the NSF GRFP under award number DGE-2139899. LFK acknowledges support by PARADIM and the Packard Foundation. Device fabrication was performed in part at the Cornell NanoScale Facility, a member of the National Nanotechnology Coordinated Infrastructure (NNCI), which is supported by the National Science Foundation (Grant NNCI-2025233). This work made use of the electron microscopy facility of the Platform for the Accelerated Realization, Analysis, and Discovery of Interface Materials (PARADIM), which is supported by the National Science Foundation under Cooperative Agreement No. DMR-2039380, and the Cornell Center for Materials Research Shared Facilities which are supported through the NSF MRSEC program (DMR-1719875). The Titan Themis 300 was acquired through NSF-MRI-1429155, with additional support from Cornell University, the Weill Institute and the Kavli Institute at Cornell.

## Author contributions

J.L.H., J.J.C., and L.F.K. conceived of the project. J.L.H., S.S., N.S., and S.D.F. performed the experiments and data analysis, with supervision from L.F.K. and J.J.C. J.L.H. wrote the manuscript, with feedback and contributions from all authors.

## Competing interests

The authors declare no competing interests.
