## [Peer Review File · Nature Communications]

In operando cryo-STEM of pulse-induced charge density wave switching in TaS₂REVIEWER COMMENTS

Reviewer #1 (Remarks to the Author):

This article systematically investigates the charge density wave (CDW) material 1T-TaS₂ exhibiting a Joule-heat-induced insulator-to-metal transition by in operando cryo-STEM experiments, which solves some of fundamental problem of resistive switching in TaS₂ devices in terms of microscopic mechanisms. This work is able to attract much attention of readers working on CDW materials. However, there are some issues in the manuscript that need to be further clarified. (1) Described in the second paragraph of the Results and Discussion “The TaS₂ CDW follows the Star-of-88 David distortion, wherein 13 Ta atoms bunch together (Fig. 1b)”. However, there are only 12 Ta atoms in Fig. 1b. (2) Due to the CDW pinning effect, is there any influence on the insulator-to-metal transition voltage under repeated pulse experiments? (3) Please describe in detail how the DNC maps are obtained? (4) It might be clearer to label each of the four images in Fig. 1f and the two images in Fig. 4b. (5) The letters labeled in the image do not match the letters in the text below in Fig.2. For example, a,b,c and A,B,C.

Reviewer #2 (Remarks to the Author):

Review of NCOMMS-23-41871-T, “In operando cryo-STEM of pulse-induced charge density wave switching in TaS₂” by James L Hart et al. This article deals with pulse-induced CDW transition of TaS₂ using a novel microscopy techniques with high temporal resolution as well as nanoscale spatial resolution. High quality experiments are being systematically assembled to clarify pulse-induced phase transition, including visualization of NC phase pinning at crystal dislocations. The content of this paper is of great interest and usefulness to a wide range of readers. The manuscript is well written and deserves publication. However, the authors should consider the remarks and questions below.

1. The authors claimed that the pulse-induced transition is driven by Joule heating. This is one of the most important findings of this study. As is well known, TaS₂ shows phase transition between the commensurate and the incommensurate phases due to temperature changes. The CDW phase transition generates periodic lattice distortion induced by the electron-phonon interaction. The authors calculated the strain from the change in lattice constants due to phase transition and then estimated the temperature. Therefore, the conclusion that pulse-induced transition is driven by Joule heating seems obvious and just a rephrasing of the already known facts. Did the authors consider factors that may cause CDW transition other than temperature?

2. The reviewer strongly encourages the authors to include error bars for strain (Figure 2c) and flake temperature (Figure 2d) to ensure the story of this paper.

3. Accuracy of the strain measurement is not sufficiently considered in this paper. The authors employed the maximum pixel intensity and center-of-mass (COM) to determine the position of the diffraction spot. A slight sample tilt from a zone axis causes an asymmetric intensity distribution as shown in Supplementary Figure 2 (and also Fig.2b). In such a case the intensity maximum will not necessarily correspond to the center position of the spot. The original pixel size seems too large compared to the length of R1 or R2 (supplementary Figure 1). These are thought to have a significant impact on the strain measurement method adopted in this study. Need additional explanation about the accuracy of strain

measurement.

4. "We then calculate the component of b^* which is perpendicular to a^* , which yields the two orthogonal in-plane lattice vectors" [Supplementary Note2]. This sentence is not readily understandable since the angle between a^* and b^* is 60 degrees as indicated in the supplemental Fig.1c.

5. "...this experiment constitutes the first in operando study to directly quantify the CDW order parameter during switching,..."[page 4, lines 9-10]. The "order parameter" as well as its definition is not specifically described in the manuscript. Need additional description.

6. "Nevertheless, the strain versus temperature data within the C phase followed a clear linear trend, both before and after the applied voltage ramps (Supplementary Figure 2)"[Supplementary Note2]. What does "linear trend" mean? Does this mean that the temperature dependence of the order parameter is linear?

7. The authors used the thermal expansion coefficient of 0.095/K. It is mandatory to assess the precision of this value to ensure the deduced temperature derived from the strain. The short description in Supplementary Note2 is not sufficient.

Reviewer #3 (Remarks to the Author):

The manuscript by Hart et al. is of great interest to the quantum materials community. Specifically, it presents the first real-time investigation of the electrically driven switching of CDW structure in 1T-TaS₂ flakes. The authors correlate the structural changes with resistivity evolution and unambiguously demonstrate that pulse-induced transitions are driven by Joule heating. Furthermore, the authors establish that dislocations have a significant impact on the switching profiles and device performance by pinning CDW domains. It is highly recommended that this manuscript be published in Nature Communications. However, there are a few questions regarding the authors' findings and their potential implications, which require further clarification.

1. For the video of temperature-induced transformation, please describe exactly where you are on the sample. Is it possible to provide images showing how large this region was?
2. Do the switching profiles remain consistent when applying consecutive pulses with the same durations and amplitudes?
2. Please report on the thickness or number of layers (approximate is fine) of the exfoliated 1T-TaS₂, especially since CDW pinning is potentially dependent on thickness.
3. Have you conducted experiments on samples with different thicknesses or capping layers that could affect the rigidity of flakes and their strain/dislocation profiles?
4. Is the prevalence of dislocations in your samples due to suspending them over a porous substrate rather than securing them to a solid one for structural support? In the manuscript, you state "We find that all exfoliated TaS₂ flakes (and many exfoliated 2D materials in general) exhibit similar dislocation structures." We suggest that you either elaborate on how many flakes you have looked at and what types or rephrase to something along the lines of "Similar dislocation structures are expected in many exfoliated 2D materials because..."
5. Figure 1c. Please add a scale bar for the diffraction patterns.
6. There are reports (e.g. Tsen, PNAS, 2015 and Masaro, Science Advances, 2015) that suggest it is possible to switch from the NC to the C state using electric biasing. Have you

observed this behavior, and do you think it is feasible based on your understanding of the Joule-driven switching mechanism?

7. Fig. 5 - Please use a different colormap or scaling. It is difficult to see the features in the maps showing response with applied bias.

Additionally, we recommend providing additional details about the experimental setup in the Methods and SI sections to enhance the reproducibility of reported experiments.

1. Provide include additional information in the Methods section about the heating chips used in your measurements. If the chips were purchased, the brand should be reported. In the case that the chips were fabricated, details of the fabrication process should be included in the Methods section.
2. Report the type/brand of the TEM holder used for the in-situ heating experiments.
3. Please report the step size between probe positions for 4D-STEM in the SI. Please also include an image of the real space probe with a scale bar in the SI.
4. Please clarify the biasing setup used to obtain data in different figures. In Figure 3, you provide a schematic of the biasing setup and report data in the normalized $V_{\text{flake}}/V_{\text{total}}$ convention. Is this normalization necessary because the current is not well-defined (constant) in the experiment? If the current is well-defined in the experiments mentioned, it would be helpful to report it for easier comparison with previous literature such as Vaskivskiy et al. (Nature Communications, 2016). Further, did you use the biasing setup illustrated in Figure 3 for experiments where you report flake resistance (e.g., Figure 4b)? To avoid confusion for the readers, please clarify the experimental setups across the different biasing experiments where you report different variables. Lastly, include the instruments used to generate the voltage outputs for different experiments to improve reproducibility.
5. Did you correct for any elliptical measurements before measuring strain, or do you only examine relative changes during switching without absolute quantification?
6. To improve reproducibility, we suggest sharing your COM algorithm and strain mapping.

We thank the reviewers for evaluating our manuscript. Based on their feedback, we believe that the revised manuscript is much stronger. Below, we provide a point-by-point response to all reviewer comments. Our responses are numbered sequentially, and in blue text. The reviewers' comments are in black. For brevity, not all changes made to the main text and Supplementary Information are reproduced in this response letter. The changes are marked in red in the revised manuscript.

Reviewer #1 (Remarks to the Author):

This article systematically investigates the charge density wave (CDW) material 1T-TaS₂ exhibiting a Joule-heat-induced insulator-to-metal transition by in operando cryo-STEM experiments, which solves some of fundamental problem of resistive switching in TaS₂ devices in terms of microscopic mechanisms. This work is able to attract much attention of readers working on CDW materials. However, there are some issues in the manuscript that need to be further clarified.

(1) Described in the second paragraph of the Results and Discussion “The TaS₂ CDW follows the Star-of-David distortion, wherein 13 Ta atoms bunch together (Fig. 1b)”. However, there are only 12 Ta atoms in Fig. 1b.

Author Response 1.1: There is one Ta atom at the center of the star, and the neighboring 12 Ta atoms are displaced towards the central atom. So there are 13 Ta atoms per cluster.

(2) Due to the CDW pinning effect, is there any influence on the insulator-to-metal transition voltage under repeated pulse experiments?

Author Response 1.2: We thank the reviewer for this insightful question. This question needs to be studied systematically and thus falls outside the scope of the current manuscript, as we feel that the flake must be switched repeatedly many times to understand how the CDW pinning affects the transition voltage. We did not carry out such systematic experiments, but below we discuss some relevant measurements from the present study.

Response Figure 1 shows the current versus voltage plots for our TaS₂ device during triangular voltage ramps. This is the same data presented in Fig. 2a of the main text, but plotted as a function of voltage. Only the ramps that produced switching are shown. These ramps were performed sequentially, and thus there is likely an increase in pinning with each measurement. As the plot shows, the switching voltage is similar for the 0.8 and 1.0 V ramps ($V_C = 0.76$ V) and slightly smaller for the 1.2 V ramp ($V_C = 0.71$ V). Hence, this data suggests that with pinning, there may be a decrease in the switching voltage. However, it would be premature to make this conclusion with this limited data, and more experiments are needed. Specifically, a series of consecutive voltage ramps with 4D-STEM mapping are needed to study the co-evolution of the CDW pinning and the switching voltage. Related to the reviewer's question is also how the density of dislocations may alter the CDW phase switching, which we are currently working on. Lastly, we note that the data in Fig. 4 of the main text provides a series of consecutive short voltage pulses that show

decrease in the resistance of the flake with repeated switching, which hints at potential pinning effects. Unfortunately, given the short pulse duration and the biasing set-up, it is difficult to extract the switching voltage from this data, and the diffraction time-resolution ($300 \mu\text{s}$) is not sufficient to capture the switching pulse ($3 \mu\text{s}$).

Thus, while we fully acknowledge and agree on the importance of the reviewer's question, it is beyond the scope of the current work as many more repeated pulse experiments are needed to draw proper conclusions. We intend to carry out these experiments as a follow-up study, which we anticipate will take several months. We hope that the reviewer understands our hesitancy in addressing this question in the current manuscript.

Response Figure 1 | Current versus voltage plots during triangular voltage ramps, with maximum voltages of 0.8, 1.0, and 1.2 V. Only the first half of the voltage ramps are shown. This is the same data displayed in Fig. 2a of the main text. For switching at 1.2 V, there is a decrease in the switching voltage.

(3) Please describe in detail how the D_{NC} maps are obtained?

Author Response 1.3: We thank the reviewer for this comment, and we have added significant text and data to better explain the D_{NC} maps. In the updated Methods and Supplementary Note 2, we provide details of the experimental parameters used for 4D-STEM mapping, as well as the processing steps used to determine D_{NC} . Additionally, we updated Supplementary Figures 6 and 7, which provide additional data to clarify how the D_{NC} maps are obtained. In the main text, we have added the following text to explain the D_{NC} maps and refer to the Supplementary for a complete explanation. Lines 267 – 271:

“With this method, the electron beam is focused to a nanoscale probe, rastered across the sample surface, and a full diffraction pattern is captured at each spatial coordinate²⁸. From the diffraction patterns, we extract the CDW angle φ and magnitude k using the same method applied to the selected area diffraction data shown in Fig. 1g,h. In turn, the D_{NC}

can be mapped in real space with a spatial resolution of ~20 nm (see Supplementary Note 2 for details).”

(4) It might be clearer to label each of the four images in Fig. 1f and the two images in Fig. 4b.

Author Response 1.4: We have relabeled Fig. 1 and Fig. 4 as suggested.

(5) The letters labeled in the image do not match the letters in the text below in Fig.2. For example, a,b,c and A,B,C.

Author Response 1.5: We thank the reviewer for catching this formatting error, and we have corrected it.

Reviewer #2 (Remarks to the Author):

Review of NCOMMS-23-41871-T, “In operando cryo-STEM of pulse-induced charge density wave switching in TaS₂” by James L Hart et al. This article deals with pulse-induced CDW transition of TaS₂ using a novel microscopy techniques with high temporal resolution as well as nanoscale spatial resolution. High quality experiments are being systematically assembled to clarify pulse-induced phase transition, including visualization of NC phase pinning at crystal dislocations. The content of this paper is of great interest and usefulness to a wide range of readers. The manuscript is well written and deserves publication. However, the authors should consider the remarks and questions below.

1. The authors claimed that the pulse-induced transition is driven by Joule heating. This is one of the most important findings of this study. As is well known, TaS₂ shows phase transition between the commensurate and the incommensurate phases due to temperature changes. The CDW phase transition generates periodic lattice distortion induced by the electron-phonon interaction. The authors calculated the strain from the change in lattice constants due to phase transition and then estimated the temperature. Therefore, the conclusion that pulse-induced transition is driven by Joule heating seems obvious and just a rephrasing of the already known facts. Did the authors consider factors that may cause CDW transition other than temperature?

Author Response 2.1: We thank the reviewer for this question. There is a misunderstanding in the reviewer’s comment which leads to the question. The reviewer states: “The authors calculated the strain from the change in lattice constants *due to phase transition* and then estimated the temperature.” To clarify, we estimate the flake temperature based on changes in lattice constant *due to thermal expansion within the C phase*. Therefore, there is no phase transition within the temperature range (and strain range) of our temperature analysis.

To expand upon this point, Response Figure 2 shows the in-plane TaS₂ lattice strain during warming. The plot shows our *in situ* TEM data (Runs 1 and 2 in blue and green), as well as data from refs. 22 and 23 of the main text. For Runs 1 and 2, between 110 and 180 K, there is a linear increase in the lattice constant owing to the thermal expansion of the C phase. Then, at ~200 K,

there is a sudden increase in the lattice constant, which is due to the transition from the C to the NC CDW phase. The strain versus temperature data from ref. 23 shows a similar response. We conclude that if the TaS₂ flake remains within the C phase, then there is a linear relationship between the strain and the temperature due to thermal expansion. As such, strain measurements can be used to determine the temperature of the C phase. Response Figure 2 has been added to the Supplementary Information.

Response Figure 2 | In-plane strain versus temperature for our TaS₂ device measured within the TEM (Runs 1 and 2), as well as data taken from the literature. The data from ref. 23 is from a capacitance dilatometer, and the data from ref. 22 is from X-ray diffraction. The ref. 22 fit is a linear fit to experimental data from 170 to 500 K; we extended the fit line down to 110 K for comparison to our experiments.

For the bias-induced CDW switching mechanism, there are two hypotheses considered in the literature. First, that the transition is purely field-induced (non-thermal), and second, that the transition is due to Joule heating (thermal). A direct experiment to differentiate these mechanisms is to study temperature changes as a function of applied bias, starting with sub-threshold voltages and increasing up to the critical voltage. For the thermal hypothesis, as the applied voltage is increased, the flake temperature should increase due to Joule heating, and at the switching voltage, the flake temperature should equal T_c . Conversely, for the non-thermal hypothesis, while temperature changes from Joule heating may be measurable, the flake temperature at switching must be less than T_c . This is the precise experiment we performed, as shown in Figure 2 of the main text. We emphasize that for voltage biases from 0.1 – 0.7 V, there is no CDW switching, and the flake remains in the C phase. Hence, for these voltages, we are in the sub-threshold regime, and the measured strain is used to calculate the change in temperature. At switching, the measured temperature equals T_c , consistent with the Joule heating hypothesis.

We have added the following sentence to the main text (lines 166 – 168) for clarity:

“Note that this method is only applicable when the flake is in the C phase (where $\Delta T = \alpha\epsilon$ is valid), since the C to NC CDW transition causes a discontinuous lattice change due to CDW-lattice coupling.”

2. The reviewer strongly encourages the authors to include error bars for strain (Figure 2c) and flake temperature (Figure 2d) to ensure the story of this paper.

Author Response 2.2: Thank you for this comment; please see author response 2.3 below for a detailed explanation. In short, the error for Fig. 2c is 0.0014%, which is now included in the Fig. 2c caption. The error bar for the temperature is now included in the updated Fig. 2d.

3. Accuracy of the strain measurement is not sufficiently considered in this paper. The authors employed the maximum pixel intensity and center-of-mass (COM) to determine the position of the diffraction spot. A slight sample tilt from a zone axis causes an asymmetric intensity distribution as shown in Supplementary Figure 2 (and also Fig.2b). In such a case the intensity maximum will not necessarily correspond to the center position of the spot. The original pixel size seems too large compared to the length of R1 or R2 (supplementary Figure 1). These are thought to have a significant impact on the strain measurement method adopted in this study. Need additional explanation about the accuracy of strain measurement.

Author Response 2.3: We thank the reviewer for this comment and agree that additional discussion of the strain is needed.

First, we note that the error of lattice constant measurements can be divided into two components, a random component σ_{rand} associated with statistical error (shot noise), and a systematic error σ_{sys} associated with sample tilt, sample position, electron optical conditions, and other variables.

If diffraction patterns are collected continuously without moving the sample or altering the electron beam conditions, then the systematic error σ_{sys} should be constant, and any variation in the measured strain will be due to statistical noise. Response Figure 3 shows strain measurements collected with the TaS₂ device at rest (*i.e.* with a fixed σ_{sys}), and the signal standard deviation is $\sigma_{\text{rand}} = 0.0014\%$.

Response Figure 3 | Measured flake strain as a function of time while the flake is held at 110 K.

To evaluate the systematic error σ_{syst} , we compare the measured lattice parameters from three separate measurements performed at 110 K within the C phase. The three measurements are separated by several hours, and in between each measurement, the TEM was switched back-and-forth between imaging and diffraction mode, and the sample was re-positioned and re-aligned. Thus, we feel comparison of these three measurements adequately represent σ_{syst} . The standard deviation of the measurements is $\sigma_{\text{syst}} = 0.031\%$.

For our experiment, we must be able to reliably measure a strain of $\sim 0.09\%$, which is the strain corresponding to a temperature difference of ~ 90 K, which is the difference between our base temperature of 110 K and the CDW transition temperature $T_c \approx 200$ K. We highlight that both σ_{rand} and σ_{syst} are significantly less than 0.09%.

For our purposes, the absolute strain value (which is primarily influenced by the systematic error) is not important. We only need to measure the change in strain as a function of temperature and applied voltage. If the flake does not substantially tilt or shift during bias, then the appropriate error for our strain measurements is σ_{rand} , and σ_{syst} is irrelevant. Response Figure 4a shows diffraction snapshots from the 0.7 V triangular ramp, taken at the beginning ($V = 0$), middle ($V = 0.7$), and end of the ramp ($V = 0$). There are no visible changes to the Bragg spot intensities, indicating that during the bias, there are no changes in the flake tilt or position. Response Figure 4b shows data from the 0.8 V ramp, at the beginning of the ramp ($V = 0$), immediately prior to the C to NC transition ($V = 0.76$), and then immediately after the C to NC transition ($V = 0.77$). While there are visible changes in the Bragg spot intensities *after* switching (indicating some degree of sample movement), we are only concerned with temperature changes within the C phase *prior* to the CDW transition. We conclude that for our strain measurements, we can accurately measure changes in relative strain which are not strongly influenced by tilt or other systematic errors.

Response Figure 4 | Diffraction patterns collected during triangular voltage ramps with maximum voltages of 0.7 V (**a**) and 0.8 V (**b**). The diffraction patterns are labeled with the applied voltage (units of volts) and measured strain. All diffraction patterns have the same intensity scaling. If the flake were tilting or moving during the voltage ramps, then the relative intensities of the Bragg peaks would change during bias. The only noticeable change in Bragg spot intensity occurs after the C to NC phase transition during the 0.8 V ramp.

To determine the sample temperature, we must know the strain state and the coefficient of thermal expansion α for TaS₂. For our device, α was determined from two separate warming cycles, one performed prior to the biasing experiments (Run 1), and one performed after the biasing experiment (Run 2). The two runs are shown below in Response Figure 5 (which is the same as Response Figure 2). We extract α based on a linear fit from 110 – 180 K, and find $\alpha = 10.3 \times 10^{-4} \% / \text{K}$ (Run 1) and $\alpha = 8.7 \times 10^{-4} \% / \text{K}$ (Run 2), with a mean value of $9.5 \times 10^{-4} \% / \text{K}$ and a standard deviation of $0.8 \times 10^{-4} \% / \text{K}$. Our results are in good agreement with bulk strain versus temperature measurements in the literature (see data from refs. 22 and 23). Note that we are looking at the temperature range of 110 – 180 K, which is below the CDW transition.

Response Figure 5 | In-plane strain versus temperature for our TaS₂ device measured within the TEM (Runs 1 and 2), as well as data taken from the literature. The data from ref. 23 is from a capacitance dilatometer, and the data from ref. 22 is from X-ray diffraction. The ref. 22 fit is a linear fit to experimental data from 170 to 500 K; we extended the fit line down to 110 K for comparison to our experiments.

In the updated manuscript, we have added error bars to Fig. 2d, which shows the flake temperature as determined from strain analysis. The error bars reflect the uncertainty in the measured value of α , which is the dominant uncertainty component. We have also adjusted the horizontal line in Fig. 2d which shows the C to NC transition temperature. The updated version reflects the uncertainty in T_c . Additionally, we have updated the Fig. 2c caption to state that the statistical error for strain analysis is $\sigma_{\text{rand}} = 0.0014\%$. We did not add error bars to Fig. 2c, since the error is comparable to the line width. Lastly, we have added discussion to Supplementary Note 3 discussing the errors for relative strain and absolute strain measurements.

Lastly, we comment on the parameters R_1 and R_2 used for COM analysis. The chosen values are optimized for our experimental data, based on careful analysis. The use of larger R_1 and R_2 values did not yield more reliable results. We note that our measured diffraction spots are very sharp, between 1 and 1.5 pixels full width at half maximum (although the spots may appear broader due to image contrast limits). Thus, a COM region with a diameter of 4 pixels (corresponding to $R_1 = 2$) is large compared to the peak width. Moreover, our use of up-sampling helps to reliably segment the data into the ‘peak’ region versus the ‘background’ region (see Supplementary Figure 6). To test the reliability and accuracy of our COM refinement, we can use the CDW angle φ , which is known to be 13.9° in the C phase. This angle provides an absolute reference to test our COM approach. Considering multiple SAED measurements of the C phase, we measure $\varphi = 13.9 \pm 0.05^\circ$ (for example, Fig. 1g of the main text and Supplementary Figure 6). This suggests a measurement error of 0.05° . In our analysis, we measure φ for N CDW spots per diffraction pattern, and then report the average value of φ . Assuming that the errors for the N diffraction spots are uncorrelated, then the uncertainty per diffraction spot is $0.05^\circ \times N^{1/2} = 0.25^\circ$, where we take $N = 25$ as a representative value. Next, we must convert the uncertainty units from degrees to pixels. To do so, we note that the scattering vector $Q_{2\text{nd}}$ used to calculate φ has a length of 10 pixels (given our

camera length), and straightforward trigonometry yields $\tan(0.25^\circ) \times 10 \text{ pixels} = 0.04 \text{ pixels}$. This represents the uncertainty projected along the vector orthogonal to $Q_{2\text{nd}}$. The full uncertainty is 0.07 pixels. Hence, our COM refinement provides deep sub-pixel accuracy, better than $1/10^{\text{th}}$ of a pixel. This is enabled by the EMPAD detector, which does not saturate and provides excellent linearity. Additionally, we use a very large electron dose, which minimizes shot noise.

4. “We then calculate the component of b^* which is perpendicular to a^* , which yields the two orthogonal in-plane lattice vectors” [Supplementary Note2]. This sentence is not readily understandable since the angle between a^* and b^* is 60 degrees as indicated in the supplemental Fig.1c.

Author Response 2.3: We have updated the text as follows:

“ a^* and b^* are not orthogonal (Supplementary Fig. 2d), so, to obtain orthogonal strain components, we extract the component of b^* which is perpendicular to a^* . This yields two orthogonal in-plane vectors.”

5. “...this experiment constitutes the first in operando study to directly quantify the CDW order parameter during switching...”[page 4, lines 9-10]. The “order parameter” as well as its definition is not specifically described in the manuscript. Need additional description.

Author Response 2.4: The order parameter is any measurable quantity which can be used to track a phase transition, and here, the order parameter is the CDW domains size D_{NC} . We have updated the text to explain this point:

Lines 104 – 106: “Here, we use D_{NC} as an order parameter to track the C to NC phase transition, where the order parameter represents a measurable quantity which distinguishes the two CDW phases; $D_{\text{NC}} \approx 10 \text{ nm}$ within the NC phase and $D_{\text{NC}} \rightarrow \infty$ within the C phase.”

Lines 137 – 139: “...this experiment constitutes the first *in operando* study to directly quantify the CDW order parameter (*i.e.* D_{NC}) during switching...”

6. “Nevertheless, the strain versus temperature data within the C phase followed a clear linear trend, both before and after the applied voltage ramps (Supplementary Figure 2)”[Supplementary Note2]. What does “linear trend” mean? Does this mean that the temperature dependence of the order parameter is linear?

Author Response 2.5: The order parameter is the CDW domain size D_{NC} , which is separate from the flake strain. In this section, we are discussing the flake strain as a function of temperature. By linear trend, we mean that the plot of strain versus temperature follows a straight line within the C phase (Please refer to Response Figure 5). We have removed the sentence mentioned by the referee, and we have rewritten Supplementary Note 2 for clarity, which is now Supplementary Note 3.

7. The authors used the thermal expansion coefficient of 0.095/K. It is mandatory to assess the precision of this value to ensure the deduced temperature derived from the strain. The short description in Supplementary Note2 is not sufficient.

Author Response 2.6: We thank the reviewer for this comment, as it is extremely important to our manuscript. Please see Responses 2.2 and 2.3 above for a detailed discussion of α . In short, we calculate the value of $\alpha = 9.5 \times 10^{-4} \% / \text{K}$ with an uncertainty of $0.8 \times 10^{-4} \% / \text{K}$. This uncertainty is now reflected in the error bars of Fig. 2d of the main text.

Reviewer #3 (Remarks to the Author):

The manuscript by Hart et al. is of great interest to the quantum materials community. Specifically, it presents the first real-time investigation of the electrically driven switching of CDW structure in 1T-TaS₂ flakes. The authors correlate the structural changes with resistivity evolution and unambiguously demonstrate that pulse-induced transitions are driven by Joule heating. Furthermore, the authors establish that dislocations have a significant impact on the switching profiles and device performance by pinning CDW domains. It is highly recommended that this manuscript be published in Nature Communications. However, there are a few questions regarding the authors' findings and their potential implications, which require further clarification.

1. For the video of temperature-induced transformation, please describe exactly where you are on the sample. Is it possible to provide images showing how large this region was?

Author Response 3.1: For all the selected-area electron diffraction experiments, we illuminate a $\sim 2 \mu\text{m}$ diameter region of the flake. The rough positioning of the beam is shown below. We chose this location since bending of the flake was minimized in this region. We have added this image to the Supplementary Information, and we state the beam size in the Methods section.

Response Figure 6 | STEM-HAADF image of the TaS₂ device, with the SAED beam position shown.

2. Do the switching profiles remain consistent when applying consecutive pulses with the same durations and amplitudes?

Author Response 3.2: We thank the reviewer for this question. This question is important and needs a systematic study with repeated switching, which we feel is outside the scope of the current study. While we did not directly study this question, preliminary results suggest that with repeated pulsing, CDW pinning reduces the switching voltage (data presented in Response Figure 1). Reviewer 1 asked a similar question, and please see our Response 1.2.

2. Please report on the thickness or number of layers (approximate is fine) of the exfoliated 1T-TaS₂, especially since CDW pinning is potentially dependent on thickness.

Author Response 3.3: The estimated sample thickness is 55 nm. This is based on the measured electron energy-loss spectroscopy thickness of 0.9 inelastic mean free paths, and an estimated mean free path of TaS₂ of 61 nm for a 120 kV beam. The estimated thickness has been added to the main text, and the methodology is described in the methods.

3. Have you conducted experiments on samples with different thicknesses or capping layers that could affect the rigidity of flakes and their strain/dislocation profiles?

Author Response 3.4: Unfortunately, with reducing flake thickness, the T_c is suppressed, and the hysteresis is widened. As such, for flakes thinner than ≈ 40 nm, we do not reliably observe a transition to the C phase with cooling down to 110 K, similar to other reports (Yoshida *et al. Sci. Adv.*, 1:e1500606, (2015)). It remains unclear why the hysteresis is widened in thinner flakes, and why the transition is fully suppressed for flakes thinner than $\approx 20 - 30$ nm. Thus, we only studied relatively thick flakes where the transition behavior is repeatable.

To avoid oxidation, h-BN capping is necessary for TaS₂. In ref. 3 of the main text (Tsen *et al. PNAS*. **112** 49 (2015)), h-BN capped TaS₂ showed the CDW transition down to 4 nm. It is not clear whether this result is related to reduced surface oxidation, or altered dislocation structures, or enhanced in-plane strain due to the capping layer (as shown by the recent work of Van Winkle, Bediako, *et al. NComms*. 14:2989 (2023)). We have not tried any capping layers for our initial study (this work) since they would complicate the biasing set-up and the diffraction analysis.

We have added the following discussion to the main text, lines 314 - 319.

“Here, we studied a relatively thick ~ 55 nm flake, and further work is needed to understand the Joule heating mechanism and dislocation pinning in thinner flakes, which show distinct CDW behavior^{3,4} and may also possess distinct defect structures³². Further work is also needed to understand interfacial effects, particularly for thin samples^{3,33}. For instance, encapsulation with hexagonal BN can strongly influence surface oxidation and strain^{3,34}, though how this influences the CDW remains poorly understood.”

4. Is the prevalence of dislocations in your samples due to suspending them over a porous substrate rather than securing them to a solid one for structural support? In the manuscript, you state “We find that all exfoliated TaS₂ flakes (and many exfoliated 2D materials in general) exhibit similar

dislocation structures.” We suggest that you either elaborate on how many flakes you have looked at and what types or rephrase to something along the lines of “Similar dislocation structures are expected in many exfoliated 2D materials because...”

Author Response 3.4: This is an important question which we are still investigating. We observe dislocations over the holes as well as over the SiN_x support, although the support makes observation of the dislocations much more difficult. Thus, we doubt that suspension over the holes is the source of the dislocations. We have updated the text as follows, lines 275 - 276:

“We find that all exfoliated TaS₂ flakes (over a dozen were observed by STEM) exhibit similar dislocation structures.”

5. Figure 1c. Please add a scale bar for the diffraction patterns.

Author Response 3.5: Thank you for pointing this out. We have added the scale bar.

6. There are reports (e.g. Tsen, PNAS, 2015 and Masaro, Science Advances, 2015) that suggest it is possible to switch from the NC to the C state using electric biasing. Have you observed this behavior, and do you think it is feasible based on your understanding of the Joule-driven switching mechanism?

Author Response 3.6: This is an excellent question, although we feel that the experimental investigation of this process is outside the scope of our manuscript. We note, however, that transitioning from the NC phase to the C phase upon cooling is dependent upon the cooling rate (Yoshida *et al. Sci. Adv.*, 1:e1500606, (2015)). Specifically, with a fast-cooling rate, the NC phase can be super-cooled and kinetically locked at low temperature. With the Joule heating mechanism, one can tailor the electric pulse to control the change in temperature as well as the cooling rate. Thus, if the base temperature is set below the nominal T_c , then short voltage pulses (with high cooling rate) could be used to switch from the C to the NC phase, and lock in the NC phase. Then, longer voltage pulses (with slower cooling rates), could be used to switching from the NC to the C phase.

The reviewer’s question and the reports the reviewer mentions need nucleation and growth studies as a function of cooling and/or heating rates either *via* the direct heating route or *via* Joule heating with voltage pulses. This would be an interesting study, but since we are speculating for now, we do not feel it is appropriate to discuss it in this manuscript.

7. Fig. 5 - Please use a different colormap or scaling. It is difficult to see the features in the maps showing response with applied bias.

Author Response 3.7: We have adjusted the scaling and updated the figure. The updated figure is shown below, for reference.

Response Figure 7 | Real-space CDW imaging during bias. **a.** Virtual STEM image which sums all of the Bragg peak intensities. The dark lines are basal dislocations. The inset is an optical image of this flake. **b.** Maps of the CDW D_{NC} as a function of applied bias. The insets show cropped diffraction patterns, extracted from local regions of 3×3 pixels. For the post-bias dataset, the top diffraction pattern is extracted from a dislocation, and the bottom diffraction pattern is extracted from a non-defective region.

Additionally, we recommend providing additional details about the experimental setup in the Methods and SI sections to enhance the reproducibility of reported experiments.

1. Provide include additional information in the Methods section about the heating chips used in your measurements. If the chips were purchased, the brand should be reported. In the case that the chips were fabricated, details of the fabrication process should be included in the Methods section.

Author Response 3.8: Thank you for this suggestion. We purchased the chips from DENS solutions (part# DS-P.T.2B4H.DS-1). We have added information regarding the chips and the experimental set-up to the Methods section.

2. Report the type/brand of the TEM holder used for the in-situ heating experiments.

Author Response 3.9: The holder was purchased from HennyZ (model FDCHB-6) and is described in detail in ref. 19 of the main text (*Microsc. Microanal.* **26**, 439–446 (2020)). We have added information regarding the holder to the Methods section.

3. Please report the step size between probe positions for 4D-STEM in the SI. Please also include an image of the real space probe with a scale bar in the SI.

Author Response 3.10: We used a step size of 21 nm. We have added this data to the Methods section. The real-space image of the probe is shown below, and has been added as Supplementary Figure 4.

Response Figure 8 | Real space image of the STEM probe used for 4D-STEM mapping, with a nominal convergence angle of 0.15 mrad.

4. Please clarify the biasing setup used to obtain data in different figures. In Figure 3, you provide a schematic of the biasing setup and report data in the normalized $V_{\text{flake}}/V_{\text{total}}$ convention. Is this normalization necessary because the current is not well-defined (constant) in the experiment? If the current is well-defined in the experiments mentioned, it would be helpful to report it for easier comparison with previous literature such as Vaskivskyi et al. (Nature Communications, 2016). Further, did you use the biasing setup illustrated in Figure 3 for experiments where you report flake resistance (e.g., Figure 4b)? To avoid confusion for the readers, please clarify the experimental setups across the different biasing experiments where you report different variables. Lastly, include the instruments used to generate the voltage outputs for different experiments to improve reproducibility.

Author Response 3.11: We thank the reviewer for this important question. For the electrical measurements shown in Figs. 1, 2, and 5 of the main text, we used a Keithley 2400 SMU to perform 2-terminal biasing measurements. For the set-up shown in Fig. 3, we used a Keysight 33600A waveform generator to produce the voltage pulses, and we used a Tektronix DPO2024 oscilloscope to monitor the V_{flake} and V_{total} signals. For the data shown in Fig. 4 of the main text, we used a similar set-up to that shown in Fig. 3, but with an additional 30Ω series resistor. The set-up is shown below in Response Figure 9. For these experiments, we applied a small DC voltage bias in addition to the pulse. By measuring the resistance drop across the 30Ω resistor with the Keithley SMU, we are then able to monitor the flake resistance on a time-scale of several seconds. Conversely, the oscilloscope only allows measurement time-scales of several μs . This was important for measuring the recovery of the flake after the pulse. This information has been added to Supplementary Note 1, which we reference within the main text in the Figure captions.

Response Figure 9 | The biasing set-up used for Figure 4 of the main text. V_{total} and V_{flake} are measured with the oscilloscope, and V_{Keithley} is measured with the Keithley SMU.

We do not have a well-controlled current source, which prevents the same type of experiment performed by Vaskivskiy *et al.* We chose to present the normalized $V_{\text{flake}} / V_{\text{total}}$ since this quantity is directly related to the device resistance, and thus highlights the switching process.

5. Did you correct for any elliptical measurements before measuring strain, or do you only examine relative changes during switching without absolute quantification?

Author Response 3.12: We did not correct for elliptical distortion, and we only consider the relative changes in strain. We have added text to emphasize the distinction between relative and absolute strain measurements in Supplementary Note 3.

6. To improve reproducibility, we suggest sharing your COM algorithm and strain mapping.

Author Response 3.13: We have uploaded the COM and strain codes to an online depository, as stated in our updated Data availability statement.

REVIEWERS' COMMENTS

Reviewer #1 (Remarks to the Author):

The authors have addressed my questions and comments. No further changes are necessary, and the manuscript can now be published as is.

Reviewer #2 (Remarks to the Author):

Having read through the revised manuscript and reply, I found that the authors answered each comment accurately and carefully, and the initial queries raised by the reviewer have been resolved. Therefore, it is of my opinion that the revised manuscript is worthy of publication.

Reviewer #3 (Remarks to the Author):

The authors have addressed all of my concerns and I have no further comments. I recommend publication in Nature Communications.

Reviewer #1 (Remarks to the Author):

The authors have addressed my questions and comments. No further changes are necessary, and the manuscript can now be published as is.

Reviewer #2 (Remarks to the Author):

Having read through the revised manuscript and reply, I found that the authors answered each comment accurately and carefully, and the initial queries raised by the reviewer have been resolved. Therefore, it is of my opinion that the revised manuscript is worthy of publication.

Reviewer #3 (Remarks to the Author):

The authors have addressed all of my concerns and I have no further comments. I recommend publication in Nature Communications.

Author response:

We thank the reviewers for their time and valuable input. We are glad that the reviewers are satisfied with our revisions.